# Genomes of the autonomous parvovirus minute virus of mice induce replication stress through RPA exhaustion

MegAnn K. Haubold[1,2,3], Jessica N. Pita Aquino[1,3,4], Sarah R. Rubin[1,3], Isabella K. Jones[1,3], Clairine I. S. Larsen[1,3,4], Edward Pham[1], Kinjal Majumder[1,2,3,4,5]*

1 Institute for Molecular Virology, University of Wisconsin-Madison, Madison, Wisconsin, United States of America, 2 Cancer Biology Graduate Program, University of Wisconsin-Madison, Madison, Wisconsin, United States of America, 3 McArdle Laboratory for Cancer Research, University of Wisconsin School of Medicine and Public Health, Madison, Wisconsin, United States of America, 4 Cell and Molecular Biology Graduate Program, University of Wisconsin-Madison, Madison, Wisconsin, United States of America, 5 University of Wisconsin Carbone Cancer Center, Madison, Wisconsin, United States of America

* kmajumder@wisc.edu

**Data Availability Statement:** Data is provided in the associated Supporting Information zip file named S1 File (RPA Paper Information Revisions).

## Abstract

The oncolytic autonomous parvovirus Minute Virus of Mice (MVM) establishes infection in the nuclear environment by usurping host DNA damage signaling proteins in the vicinity of cellular DNA break sites. MVM replication induces a global cellular DNA Damage Response (DDR) that is dependent on signaling by the ATM kinase and inactivates the cellular ATR-kinase pathway. However, the mechanism of how MVM generates cellular DNA breaks remains unknown. Using single molecule DNA Fiber Analysis, we have discovered that MVM infection leads to a shortening of host replication forks as infection progresses, as well as induction of replication stress prior to the initiation of virus replication. Ectopically expressed viral non-structural proteins NS1 and NS2 are sufficient to cause host-cell replication stress, as is the presence of UV-inactivated non-replicative MVM genomes. The host single-stranded DNA binding protein Replication Protein A (RPA) associates with the UV-inactivated MVM genomes, suggesting MVM genomes might serve as a sink for cellular stores of RPA. Overexpressing RPA in host cells prior to UV-MVM infection rescues DNA fiber lengths and increases MVM replication, confirming that MVM genomes deplete RPA stores to cause replication stress. Together, these results indicate that parvovirus genomes induce replication stress through RPA exhaustion, rendering the host genome vulnerable to additional DNA breaks.

## Author summary

Parvoviruses are used in the clinic to design recombinant gene therapy vectors and as oncolytic agents. The autonomous parvovirus MVM utilizes the host cell's DNA damage response machinery to replicate in host cells and cause additional DNA breaks. However, the mechanism of MVM-induced DNA damage remains unknown. We have discovered that MVM sequesters the host DNA repair protein RPA, which normally associates with

**Funding:** This study was supported by a National Institute of Allergy and Infectious Diseases grant AI148511 to Dr. KM, a National Science Foundation grant 2018238648 and a University of Wisconsin-Madison SciMedGRS Graduate research fellowship to JNPA, a University of Wisconsin-Madison Hilldale Fellowship to SRR, a University of Wisconsin-Madison Sophomore Research Fellowship to IKJ. This study was also supported by a University of Wisconsin-Madison Office of the Vice Chancellor for Research and Graduate Education Start-up support, a University of Wisconsin-Madison School of Medicine and Public Health Start-up support, and a University of Wisconsin Carbone Cancer Center Start-up support to Dr. KM, plus by National Institute of Health grant P30 CA014520. The funders had no role in study design, data collection and analysis, decision to publish, or preparation of the manuscript.

**Competing interests:** The authors declare no competing financial interest.

single stranded DNA in the nucleus, rendering the host genome susceptible to replication stress. Our study provides insights into the mechanisms utilized by single-stranded DNA viruses to amplify host-cell DNA damage.

## Introduction

Parvoviruses are small, non-enveloped, single-stranded DNA viruses with a linear genome that are used as gene therapy vectors and oncolytic agents in the clinic [1–3]. Unlike Adeno-Associated Viruses (AAV) that are used to design gene therapies and require a coinfecting helper virus (Adenovirus, Herpesvirus or Papillomavirus) to replicate, the oncolytic protoparvovirus Minute Virus of Mice (MVM) replicates autonomously in host cells in S phase [4–6]. This replication is carried out by an alternating strand displacement and rolling circle pathway referred to as "rolling hairpin" replication, that originates at the palindromic Inverted Terminal Repeats (ITRs) flanking the 5 kb viral genome [5]. Since it does not encode its own polymerases, MVM is completely dependent on host replication proteins, primarily pol-alpha and delta, to amplify its genome [6]. MVM replication critically depends on its non-structural protein NS1 that transcriptionally activates the expression of the viral capsid genes, functions as a helicase and a nickase to cleave dimer MVM genomes during replication and transports the viral genome to cellular sites of DNA damage [7,8]. The other non-structural protein encoded by MVM, NS2, is only required for replication in mice but not in transformed human cells and functions in nuclear export in murine cells [9].

Infection of host cells by DNA viruses elicit nuclear DNA Damage Response (DDR) signals that mirror those initiated upon induction of cellular DNA breaks [10,11]. Designed to maintain the fidelity of the host genome from genotoxic stress, these cellular DDR signals can impede or facilitate the life cycle of viral pathogens [12]. Therefore, viruses have evolved distinct mechanisms to modulate the host DDR pathways, as is the case for Adenovirus, that degrades the DDR protein MRE11 using its oncoproteins E1B55K and E4ORF6 to overcome the suppressive effect of the host DDR [13–16]. SV40 and Polyomaviruses on the other hand utilize the host DDR pathways for their benefit through their Large T Antigen (LTA) proteins, that interact with DNA damage signaling proteins to enhance their ability to replicate [17–19]. We have previously discovered that MVM utilizes its non-structural protein NS1 to bind and localize the viral genome to pre-existing cellular sites of DNA damage that contain host replication and repair proteins, presumably to initiate virus replication [8,20]. MVM establishes replication centers at this nuclear milieu, forming virus replication factories dubbed APAR (Autonomous Parvovirus Associated Replication) bodies. APAR bodies coincide with the viral genome and non-structural proteins NS1 and NS2; as well as host replication and DNA damage signaling proteins such as DNA polymerase delta, MRE11, RPA and many more [6,21–25]. Although MVM replication requires signals from DNA damage signaling proteins in the ATM kinase pathway, at late stages it inactivates the ATR signaling pathway, the cell's primary responder to single-stranded DNA breaks generated by replication stress [21,22,26]. It is speculated that inactivation of the ATR signaling pathway serves to inactivate the proteins responsible for detecting single-stranded MVM genomes, thus facilitating viral pathogenesis [21]. MVM infection eventually leads to a potent pre-mitotic cell cycle block at the G2/M border mediated by degradation of cellular p21 and transcriptional repression of Cyclin B1 during which the host genome is also fragmented [27–30].

MVM genomes localize to cellular sites that are rich in DDR proteins even in the absence of viral infection. Chemical induction of replication stress using hydroxyurea (HU) preferentially

enriches DDR proteins at these same cellular sites [20]. We have previously dubbed these cellular sites where MVM establishes replication centers in the nuclear milieu as Virus Associated Domains (VADs), which also colocalize with Early Replicating Fragile (ERF) sites, where transcription-replication conflicts take place at higher frequency leading to a high basal level of DNA damage signals [20,31,32]. Interestingly, in addition to transporting the viral genomes to the cellular VAD sites, ectopically expressed NS1 proteins also associate with cellular VADs [8]. As virus replication amplifies, MVM genomes spread along the host genome to induce further DNA damage beyond the VADs, causing the recruitment of additional DNA damage signaling proteins. Induction of additional cellular replication stress leads to enhanced MVM replication, suggesting a causal link exists between replication stress genome-wide and virus pathogenesis [20].

MVM infection inactivates signaling through the ATR-CHK1 kinase pathway, responsible for maintaining host genome integrity in response to replication stress [21,26]. ATR activation is initiated by the binding and autophosphorylation of single-stranded DNA molecules by Replication Protein A (RPA). The ssDNA-RPA subunits sequentially recruit ATRIP, the checkpoint clamp complex RAD9-HUS1-RAD1 (9-1-1) and TOPBP1, leading to phosphorylation of ATR [26]. MVM infection inactivates the ATR pathway at the level of TOPBP1, although earlier members of the ATR signaling cascade continue to associate with APAR bodies [21]. In doing so, MVM inactivates the downstream CHK1 kinase, normally required to counter new origin firing in response to replication stress. However, it remains unknown how MVM induces replication stress in host cells and whether this is exacerbated by ATR-CHK1 inhibition. Since parvoviral replication utilizes many host DNA damage signaling proteins that are normally at cellular replication forks, it additionally remains possible that stoichiometric competition for host DNA damage signaling proteins exacerbates replication fork instability and eventual fork collapse. Some of these host factors usurped by MVM could include those that that have been historically found associated with MVM APAR bodies, including RPA and PCNA, which are essential for MVM replication and associate with NS1 during ectopic expression [20,25,33].

In this study, we show that MVM infection starts to induce host replication stress during early S phase soon after initiation of virus replication, even when infected with replication incompetent UV-inactivated MVM. In addition, MVM induces replication stress by modulating host origin firing, as well as by serving as a sink for cellular RPA molecules. MVM-induced replication stress and origin misfiring can be rescued by inhibiting CDC7 activity, as well as by overexpressing exogenous RPA during infection. Together, these results indicate that MVM genomes induce cellular replication stress through RPA exhaustion, contributing host genome instability and induction of cellular DNA breaks.

## Results

### MVM infection induces cellular replication stress that precede DNA damage signals

To determine how MVM infection induces cellular DNA damage, we performed single-molecule DNA Fiber Analysis (DFA) in para-synchronized murine A9 cells infected at high multiplicity of infection (MOI of 25) with the prototype strain MVMp. DFA experiments were performed by sequential pulses of IdU and CldU for 20 minutes each at the indicated timepoints of MVM infection prior to slide fixation (Fig 1A). In our synchronization protocol, the host cells enter S phase approximately 10 hours post release (and post infection; S1 Fig). As shown in Fig 1B and 1C, the lengths of the DNA fibers in cells pulsed with either IdU or CldU were decreased in MVM-infected samples at 12 hours post-infection (hpi) when compared

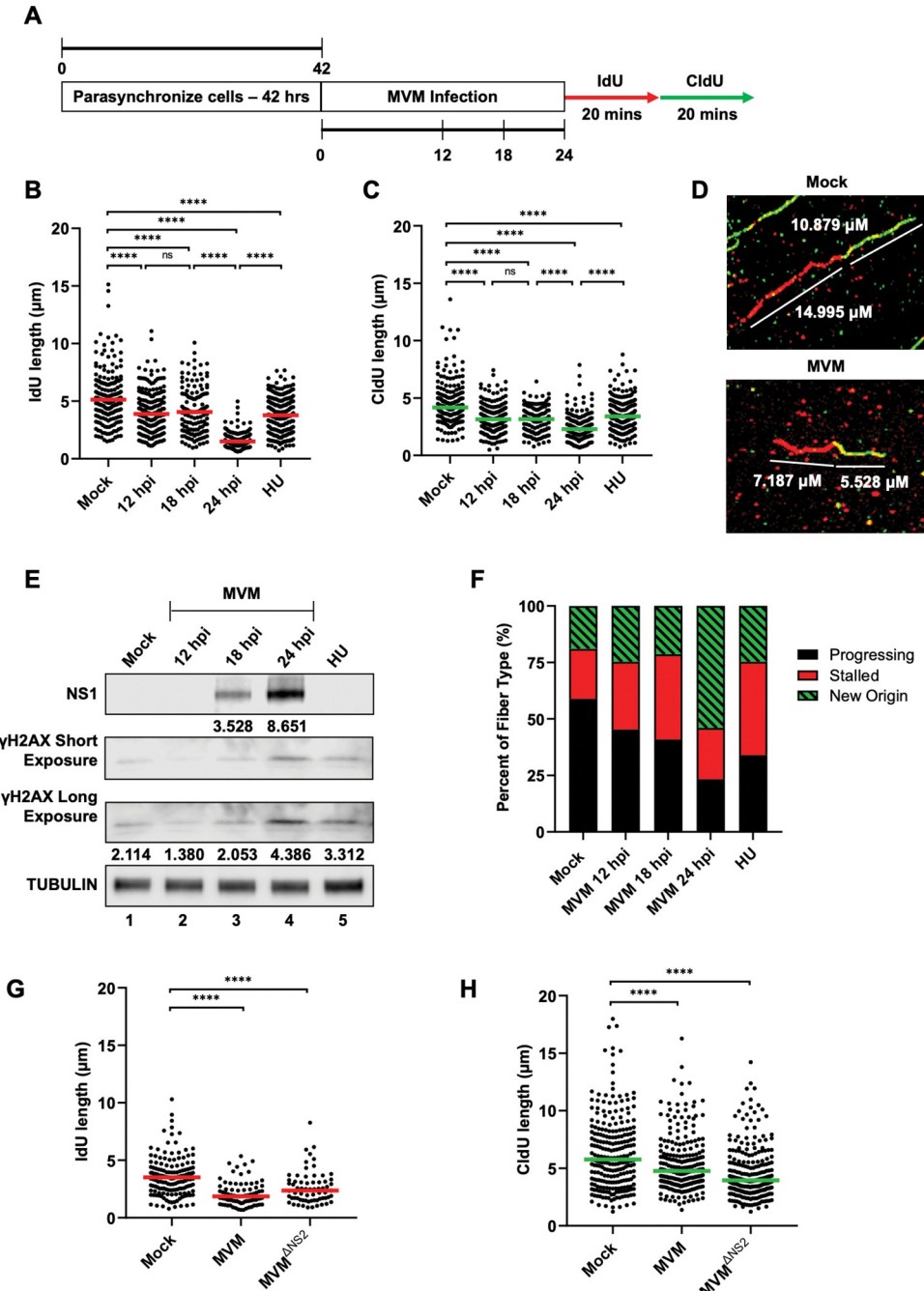

**Fig 1. MVM infection induces cellular replication stress that precede DNA damage signals.** (A) Schematic of the timeline of Single-Molecule DNA Fiber Assays during MVM infection. Mouse A9 fibroblasts are synchronized in Isoleucine deficient media for 42 hours before being released into complete DMEM and concurrently infected with MVMp at an MOI of 25 or treated with hydroxyurea (HU) at a concentration of 2μM. At the indicated timepoints post-infection (12, 18, and 24 hours post infection), cells were sequentially pulsed with IdU and CldU and processed for DFA. (B, C) Each datapoint represents the length of a single IdU and/or CldU labelled DNA fiber. The median length of many measurements of IdU and CldU labelled fibers are represented by red and green horizontal bars respectively. The experiment was performed as described by the schematic depicted in (A). At least 150 individual fibers were measured for each condition. Similar results were obtained for three independent biological replicates of MVM infection. Statistical significance was determined by Mann Whitney Wilcoxon test, **** represents P ≤ 0.0001, ns represents not statistically significant. (D) Representative DFA images of single fibers in Mock infected A9 cells (top panel) compared with MVM infected A9 cells at 24 hpi (bottom panel) with the respective measurements of the IdU and CldU lengths indicated in white text. (E) Western blot showing the levels of NS1 and γH2AX for viral replication

and DNA damage, respectively, over the time-course of MVM infection. The respective band intensities are shown below the sample. The top row of γH2AX is taken with a 10 minute exposure (short) and the bottom with a 20 minute exposure (long). Cells pulsed with Hydroxyurea at 2μM concentration is a positive control for DNA damage, as has been previously described [22]. Tubulin serves as the loading control for the Western blots. (F) Categorization of DNA fiber types as percentages out of a total of 100% as determined by presence of IdU or CldU, divided into percentages that are progressing replication forks (black), stalled replication forks (red) and new origins (green with black stripes). (G,H) Non-synchronous U2OS cells were infected with wild-type MVMp or the NS2-deficient (MVM$^{ΔNS2}$) mutant of MVM at an MOI of 25 for 24 hours before being pulsed with IdU/CldU as described in the schematic in (A), and processed for imaging. At least 150 individual fibers were measured for each condition. Similar results were obtained for three independent biological replicates of MVM infection. Statistical significance was determined by Mann Whitney Wilcoxon test, **** represents P ≤ 0.0001.

with uninfected cells which were in the same stage of the cell cycle (i.e. early S phase). As replication progressed, we observed a progressive shortening of replication fibers at 18 hpi and 24 hpi, comparable to that of fibers undergoing HU induced replication stress (Fig 1B and 1C; representative fibers shown in Fig 1D). To independently corroborate the impact of replication fiber shortening with induction of cellular DNA damage signals, we measured the cellular levels of phosphorylated H2AX (γH2AX) alongside those of the viral non-structural phosphoproteins NS1, a marker for virus replication (Fig 1E). Interestingly, although we had observed a shortening of host replication fibers at 12 hpi, we did not detect significant levels of γH2AX at this timepoint (Figs 1E, lane 2 and S2, lane 1). This observation suggested that replication stress precedes the induction of cellular DDR signals during MVM infection. We categorized the DNA fibers associated with replication forks at different stages over the time-course of MVM infection, observing a large percentage of new origins firing at 24 hpi (Figs 1F; green fractions and schematized in S3). Since MVM infected cells at 24 hpi are in G2 phase of cell cycle [22,28,30] during which new origins are not expected to fire, this suggested that MVM infection may modulate the machinery used by the host cell to control replication origin firing. To determine the generalizability of our findings of MVM-induced replication stress, we measured cellular DNA fibers in U2OS cells infected with MVM and the NS2-expression mutant of MVM (MVM$^{ΔNS2}$; verified in S4A Fig) at 24 hpi, which infects only transformed human cells. These measurements revealed that both wild-type MVM and MVM$^{ΔNS2}$ infection induce replication stress at significant levels relative to mock in U2OS cells (Fig 1G and 1H). Additionally, MVM$^{ΔNS2}$ generated the same proportion of replication fork events in infected U2OS cells as wild-type MVM (S4B Fig). Taken together, our studies on the impact of MVM infection in A9 and U2OS cells revealed that MVM induces replication stress that precedes the induction of cellular DDR, leading to increased new-origin firing at late stages of infection.

## MVM-induced replication stress is reversible by CDC7 inhibition

Licensing of the host replication complex during DNA replication is carried out by phosphorylation of the Minichromosome Maintenance (MCM) helicase complex by the serine/threonine kinase CDC7 [34]. The ATR-CHK1 pathway stabilizes CDC7 during replication stress to prevent replication fork collapse [35]. However, MVM infection inactivates the ATR/CHK1 pathway at late stages, suggesting that early in infection (12 hpi and 18 hpi) there is sufficient nuclear CHK1 to respond to replication stress and unscheduled origin firing. However, we saw major dysregulation of new origin firing during the late stages of MVM infection (Fig 1F). Therefore, we hypothesized that ATR/CHK1 inactivation at late stages of infection dysregulates CDC7 and its ability to regulate new origin firing. To test this hypothesis, we treated MVM infected A9 cells with the CDC7 inhibitor PHA 767491 at 20 hpi (iCDC7; as illustrated in Fig 2A), which led to the rescue of DNA fiber shortening at 24 hpi (Fig 2B and 2C), correlating with a decrease in new origin firing in the host cells (Fig 2D). In addition, treating with the

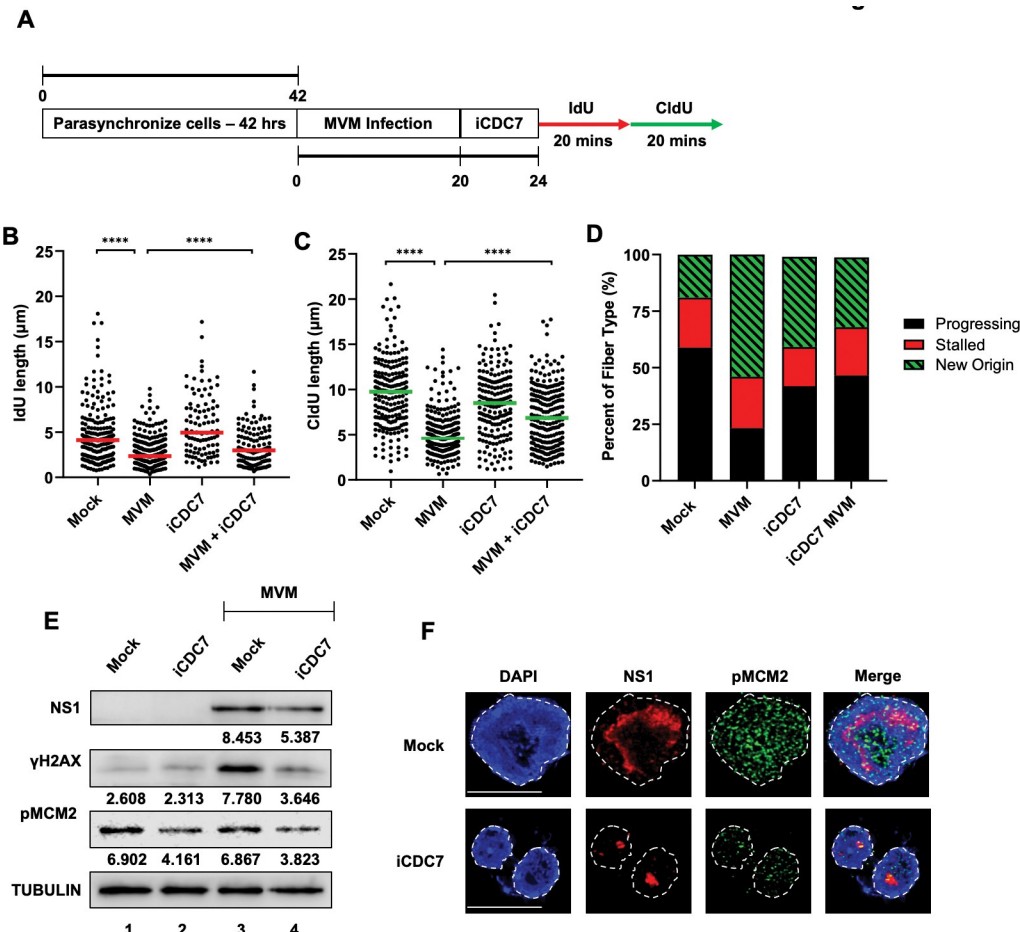

**Fig 2. MVM-induced replication stress is reversible by CDC7 inhibition.** (A) Schematic of the Single-Molecule DNA Fiber Assays using the CDC7 inhibitor PHA 767491 (Millipore Sigma). Mouse A9 fibroblasts were synchronized in Isoleucine deficient media for 42 hours before being released into complete DMEM media and concurrently infected with MVMp at an MOI of 25. At the indicated 20 hpi timepoint, 5μM of iCDC7 was added to the cells for 4 hours. At 24 hpi, MVM infected iCDC7 pulsed cells were sequentially pulsed with IdU and CldU for 20 minutes each before being processed for DFA analysis. (B, C) Each datapoint represents the length of a single IdU and/or CldU labelled DNA fiber. The median length of many measurements of IdU and CldU labelled fibers are represented by red and green horizontal bars respectively. The experiment was performed as described by the schematic depicted in (A). At least 150 individual fibers were measured for each condition. Similar results were obtained for three independent biological replicates of MVM infection. Statistical significance was determined by Mann Whitney Wilcoxon test, **** represents P ≤ 0.0001. (D) Categorization of DNA fiber types as percentages out of a total of 100% as determined by presence of IdU or CldU, divided into percentages that are progressing replication forks (black), stalled replication forks (red) and new origins (green with black stripes). (E) Western blot showing the levels of NS1 and γH2AX for viral replication and cellular DNA damage, respectively, upon pulsing with iCDC7 as shown in the schematic in (A) prior to MVM infection. Activity of the CDC7 inhibitor activity was verified by phospho-MCM2 levels and Tubulin was served as loading control for the western blots. The respective band intensities are shown below the sample. (F) Samples from lanes 3 and 4 in panel (E) were processed for formation of viral replication centers by NS1 staining (red) and phospho MCM2 levels (green) measured for CDC7 inhibitor activity. Cells infected with MVM at 24 hpi under mock-treated conditional (top panel) were compared to those pulsed with iCDC7 (bottom panel) by NS1 staining (red). White broken line demarcates the nuclear border, which was identified by DAPI staining (blue). The white scalebar represents 10 microns.

CDC7 inhibitor (which decreased phosphorylation of the MCM2 helicase, Fig 2E) resulted in lowered levels of NS1 and γH2AX in MVM infected A9 cells, as detected via western blot (Fig 2E, compare lanes 3 and 4). The diminished levels of γH2AX in the presence of iCDC7 support the rescue of replication fiber length observed in Fig 2B and 2C, indicating lower levels of

DNA damage and replication stress in the host cell. This decrease of MVM replication was further corroborated by smaller nuclear APAR bodies monitored by NS1 staining in MVM-infected A9 cells at 24 hpi (Fig 2F). Taken together, these findings suggested that MVM infection at late stages of infection induces replication stress through the dysregulation of the replication kinase CDC7.

## Ectopically expressed MVM Non-structural proteins NS1 and NS2 cause replication fork shortening

Characterization of the cellular DDR during MVM infection has previously determined that the MVM-induced DNA damage signals require ongoing virus replication [22]. Since host cell replication stress precedes virus replication, we attempted to determine which viral elements are sufficient to induce host-cell replication stress. To assess the contribution of MVM non-structural proteins to host-cell replication stress, we transfected plasmids expressing NS1 and NS2 into A9 cells [which have been previously shown to not induce cellular γH2AX; [22]] and assessed their impact on host replication forks by DNA Fiber Assays at 24 hours post-transfection (Fig 3A). Ectopic expression of both NS1 and NS2 were sufficient to induce shortening of host-cell replication forks, observed by decreasing length of both IdU and CldU (Fig 3B and 3C). Since our observation of NS1-induced replication stress might be generated downstream of cytopathic cellular effects, we overexpressed an ATP-ase mutant of NS1 which is incapable of binding to DNA [labelled as NS1$^{K405S}$; [36,37]]. Overexpression of NS1$^{K405S}$ in A9 cells did not lead to an impact on host-cell replication forks, suggesting the DNA binding function of NS1 is sufficient to induce host cell replication stress (Fig 3B and 3C; NS1$^{K405S}$ samples). Although the NS1 overexpression did not impact the types of replication fibers at 24 hours post-transfection, NS2 overexpression (also for 24 hours) led to a slight increase in stalled replication forks in addition to shortening of both IdU and CldU tracks (Fig 3B–3D). Taken together, these results suggested that individual MVM proteins are sufficient to induce replication stress without the induction of additional cellular DNA damage signals.

## The non-replicating MVM genome binds cellular RPA and induces replication stress

Since MVM infection induced replication stress in host cells entering S phase at the earliest timepoints prior to detectable NS1 (Fig 1B, 1C and 1E; 12 hpi timepoints) we reasoned that NS1 might not be sufficient to cause replication stress in host cells early in infection. Therefore, we examined the ability of the non-replicating MVM genome to induce replication stress in the host using UV-inactivated MVM, which has been previously shown to not induce cellular γH2AX [[22], S5A Fig]. As shown in Fig 4B and 4C, UV-MVM infection was sufficient to cause shortening of host replication forks, comparable to that of replicating MVM. Categorization of the replication fiber types revealed that UV-MVM genomes induced new origin firing in the same way as productive MVM infection (Fig 4D). These findings with UV-inactivated MVM genomes indicated to us that the mere presence of the ssDNA viral genome is enough to induce replication stress on the host cell, as well as dysregulate new origin firing. Because the UV-MVM genome is largely made up of ssDNA [[22], S5B Fig], we hypothesized that the presence of the viral genome may serve as a sink for RPA molecules in the cell, preventing it from properly protecting the host genome, leaving it vulnerable to replication stress-induced fork collapse. Consistent with this prediction, during a timecourse of MVM infection there was a progressive increase of nuclear phospho-RPA that colocalized with APAR bodies (S6 Fig). To determine if RPA binds to the replication incompetent UV-MVM genome, we performed ChIP-qPCR on the MVM genome on the P4 promoter, finding that RPA associated

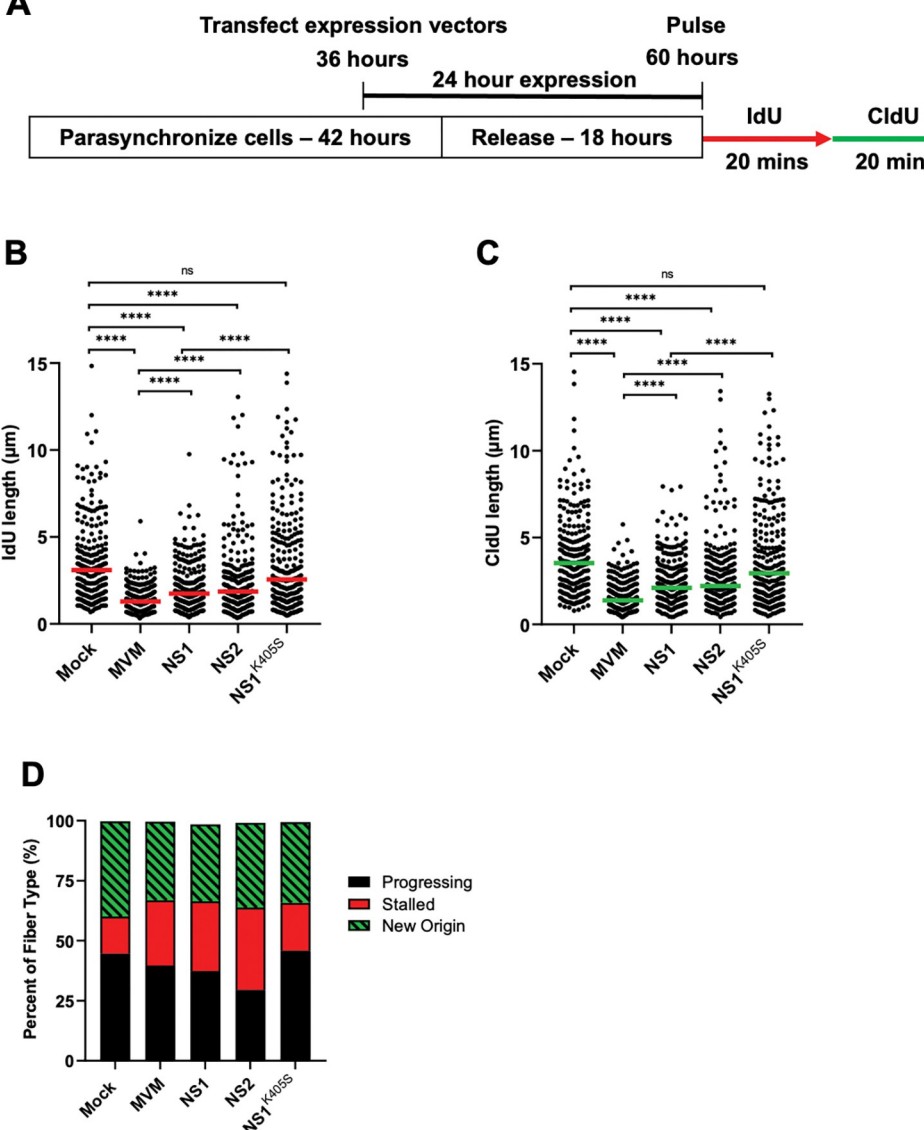

**Fig 3. Ectopically expressed MVM Non-structural proteins NS1 and NS2 cause replication fork shortening.** (A) Schematic of the timeline of the Single-Molecule DNA Fiber Assays during ectopic expression of MVM non-structural proteins in parasynchronized A9 cells. Mouse A9 fibroblasts were transfected with 0.5 µg of pUC18 (labelled as Mock) and NS1, NS2, or NS1$^{K405}$ expression vectors using Lipo293D transfection reagent during parasynchronization or infected with MVM at an MOI of 25 for 24 hours. At 24 hours post-transfection, transfected cells were sequentially pulsed with IdU and CldU for 20 minutes each before being processed for DFA analysis. (B, C) Each datapoint represents the length of a single IdU and/or CldU labelled DNA fiber. The median length of at least 150 measurements of IdU and CldU labelled fibers from each treatment are represented by red and green horizontal bars respectively. The experiment was performed as described by the schematic depicted in (A). At least 150 individual fibers were measured for each condition. Similar results were obtained for three independent biological replicates of MVM infection. Statistical significance of the data was determined by Mann Whitney Wilcoxon test, **** represents P ≤ 0.0001, ns represents not statistically significant. (D) Categorization of DNA fiber types as percentages out of a total of 100% as determined by presence of IdU or CldU, divided into percentages that are progressing replication forks (black), stalled replication forks (red) and new origins (green with black stripes).

with the UV-inactivated MVM genome at substantially higher levels than non-specific IgG (Fig 4E). Since cellular replication stress generates single stranded DNA at host replication forks that are subsequently coated with RPA, we imaged the location of RPA molecules relative

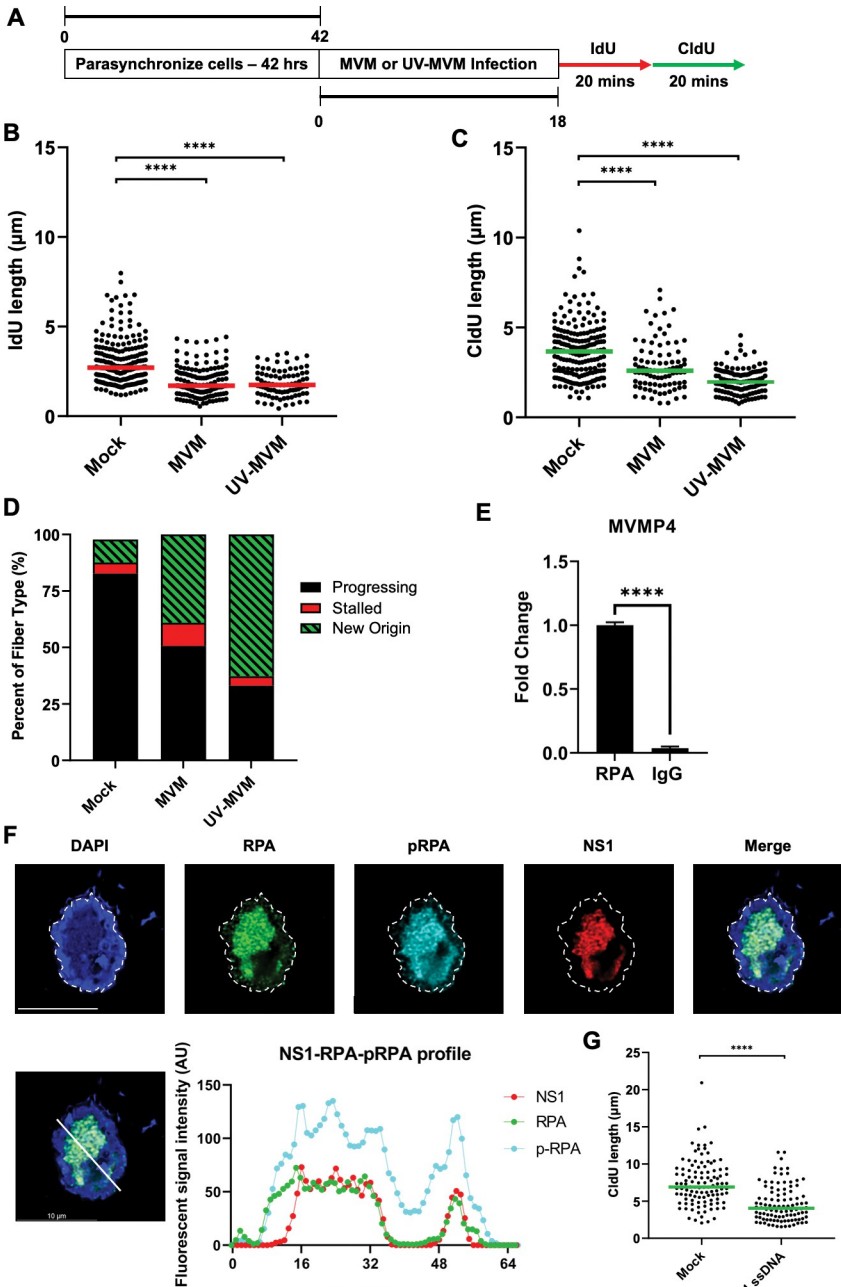

**Fig 4. The non-replicating MVM genome binds cellular RPA and induces replication stress.** (A) Schematic of a Single-Molecule DNA Fiber Assay timeline. Mouse A9 fibroblasts were synchronized in Isoleucine deficient media for 42 hours, then released into complete media and infected with MVM or UV-MVM at an MOI of 25 for 18 hours. At 18 hours post-infection, cells were sequentially pulsed with IdU and CldU for 20 minutes each before being processed for DFA analysis. (B, C) Each datapoint represents the length of a single IdU and/or CldU labelled DNA fiber. The median length of at least 150 measurements of IdU and CldU tracks are represented by red and green horizontal bars respectively. The experiment was performed as described by the schematic depicted in (A). At least 150 individual fibers were measured for each condition. Similar results were obtained for three independent biological replicates of MVM infection. Statistical significance of the data was determined by Mann Whitney Wilcoxon test, **** represents P ≤ 0.0001. (D) Categorization of DNA fiber types as percentages out of a total of 100% as determined by presence of IdU or CldU, divided into percentages that are progressing replication forks (black), stalled replication forks (red) and new origins (green with black stripes). (E) ChIP-qPCR of RPA binding to the UV-MVM genome at 18 hpi assessed at the MVMP4 promoter compared with IgG as the background control. Measurements are represented as percent pulldown of input chromatin, **** represents P ≤ 0.0001. (F) Immunofluorescence analysis of MVM infected A9 cells at 18 hpi stained with antibodies detecting NS1 (red), total RPA (green) and DNA damage markers phosphorylated

RPA (cyan). The nuclear boundary is demarcated by white dotted line and the nucleus is stained with DAPI (blue). The white scalebar represents 10 microns. Fluorescent intensity depicting colocalization of NS1, total RPA and phosphorylated RPA is shown across a lateral section of a representative MVM infected nucleus in the lower left. (G) Each datapoint represents the length of a single CldU labelled DNA fiber. The median length of at least 100 measurements of CldU tracks are represented by green horizontal bar. The experiment was performed as described by the timeline in (3A) with small oligos. At least 100 individual fibers were measured for each condition. Similar results were obtained for three independent biological replicates of MVM infection. Statistical significance of the data was determined by Mann Whitney Wilcoxon test, **** represents P ≤ 0.0001.

to APAR bodies. Strikingly, we found that both total RPA and phospho-RPA molecules localized within MVM replication centers (Fig 4F). To confirm that the single stranded MVM genome is sufficient to induce replication stress in the host, we designed small ssDNA oligos containing a portion of the MVM promoter 4 (P4) and transfected them (Fig 3A) into A9 cells. Compared with the double stranded pUC18 vector, the P4 oligo was sufficient to induce shortening of replication fibers upon transfection (Fig 4G). Taken together, these findings suggested that the MVM genome might be a substrate for RPA exhaustion, depleting the cellular stores of this critical single-stranded DNA binding protein and rendering the host genome vulnerable to replication stress.

### RPA overexpression rescues MVM-induced replication stress

To confirm that depletion of cellular RPA levels leads to MVM-induced replication stress, we overexpressed phospho-RPA32 ectopically by transient transfection in A9 cells during para-synchronization [designated as pRPA [38] and schematized in Fig 5A]. Upon infection with UV-inactivated MVM, the presence of extra molecules of phospho-RPA in the nuclear compartment was sufficient to rescue host replication fibers, as measured by both IdU and CldU lengths (Fig 5B and 5C). However, overexpression of phospho-RPA in host cells did not impact the fraction of DNA fiber types in host cells (Fig 5D) and led to a modest increase in MVM replication (Fig 5E). Taken together, our findings indicate that the MVM genome serves as a sink for cellular stores of RPA to induce replication stress in host cells.

### Discussion

In this study, we show for the first time using single-molecule DNA fiber assays that single-stranded DNA viruses induce replication stress in host cells that precedes virus replication and virus-induced DNA damage signals. RPA binds to the UV inactivated MVM genomes and the overexpression of phosphorylated RPA32 molecules leads to rescue of replication fork shortening during MVM infection. Ectopic expression of NS1 and NS2 phosphoproteins are sufficient to induce replication stress in the host genome. We therefore propose that exhaustion of cellular RPA levels contribute to virus-induced host genome instability at the early stages or infection, which is further amplified by the presence of NS1, NS2 and more replicating MVM molecules in the nuclear environment. As MVM replication continues, at 24 hpi (late stages of viral infection), MVM induces aberrantly high firing of replication origins through the CDC7 kinase. Consistent with this finding, CDC7 inhibition rescued host fibers and their propensity to fire replication origins. Taken together, these findings suggest that the viral genome serves as a sink for cellular stores of RPA, causing misfiring of cellular replication forks and contributing to host genome instability which is beneficial for the virus life cycle.

Characterization of the interplay between MVM infection and cellular DNA damage signals have previously shown that MVM induces cellular DNA damage responses that are detectable by temporal increase of cellular and local γH2AX as well as the presence of comet tails indicative of genome fragmentation [20,22]. Interestingly, pulsing MVM infected cells with the

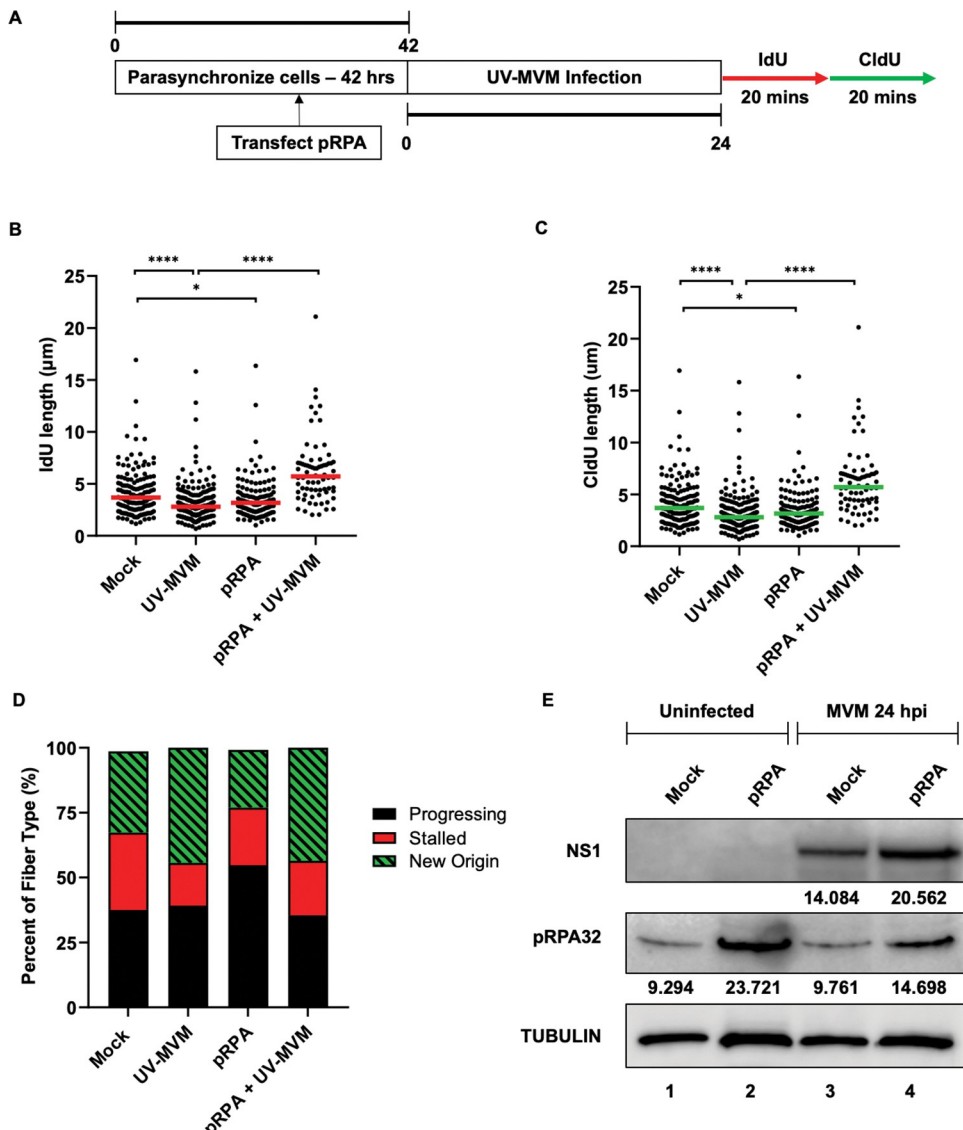

**Fig 5. RPA overexpression rescues MVM-induced replication stress.** (A) Schematic of the Single-Molecule DNA Fiber Assay where RPA is overexpressed in A9 cells prior to infection. Mouse A9 fibroblasts were synchronized in Isoleucine deficient media during which they were transfected with 1 microgram of pUC18 (labelled Mock) or an RPA expression vector at 12 hours post-initiation of synchronization. Cells were released into complete DMEM media at 42 hours when they were concurrently infected with UV-MVM at an MOI of 25. Cells were then infected with UV-MVM at an MOI of 25 for 24 hours. Cells were pulsed with IdU and CldU before completing the rest of the DFA protocol. (B, C) Each datapoint represents the length of a single IdU and/or CldU labelled DNA fiber. The median length of at least 150 measurements of IdU and CldU tracks are represented by red and green horizontal bars respectively. The experiment was performed as described by the schematic depicted in (A). At least 150 individual fibers were measured for each condition. Similar results were obtained for three independent biological replicates of MVM infection. Statistical significance of the dataset was determined by Mann Whitney Wilcoxon test, **** represents $P \leq 0.0001$, * represents $P \leq 0.05$. (D) Categorization of DNA fiber types as percentages out of a total of 100% as determined by presence of IdU or CldU, divided into percentages that are progressing replication forks (black), stalled replication forks (red) and new origins (green with black stripes). (E) Western blot analysis of MVM infection during ectopic expression of RPA32 showing cellular levels of NS1 as a marker for virus replication, phospho-RPA32 levels due to expression and induction of cellular DNA damage and total Tubulin levels in the cell as loading control for the immunoblot. The respective band intensities are shown below the sample.

replication stress inducing agent hydroxyurea (HU) led to increased virus replication [20]. These observations initially suggested that HU-induces the formation of more DNA break sites on the host genome which could serve as potential sites for APAR body formation [8,20]. Our findings in this study build on the proviral nature of host cell replication stress by eluci-dating the molecular events that drive the cellular DNA break formation at replication forks. Since we detect MVM induced replication stress as early as 12 hours post-release when MVM genomes are yet to amplify vigorously, this suggests that the mere presence of the MVM genome in small quantities can induce replication stress on the host cell. These findings are further borne out by induction of cellular replication stress even in the absence of MVM repli-cation when infected with UV-inactivated MVM genomes which do not induce cellular γH2AX [22]. We propose that this cellular stress is generated via multiple pathways that func-tion at different stages of MVM infection. Prior studies on the late stages of MVM infection (at 24 hpi and beyond) have identified CRL4$^{Cdt2}$- mediated p21 degradation and transcriptional repression of Cyclin B1 as key events driving cell cycle arrest at the late stages of infection [27–30]. Our findings indicate RPA exhaustion as one of the additional host stress pathways which is activated at early stages of MVM infection.

Upon replication stress, the ATR signaling cascade is initiated by RPA binding and subse-quent phosphorylation on single stranded DNA. In doing so, single-stranded DNA molecules in the nuclear environment serve as the "lesions" that have driven the need for the evolution of the ATR signaling pathway. The phosphorylated RPA (pRPA) molecules recruit ATRIP and ATR proteins which signal two parallel downstream signaling pathways- (i) mediated by TOPBP1 and RAD9-RAD1-HUS1 complex and (ii) through the protein ETAA1 to autoregu-late ATR phosphorylation [39]. Interestingly, ETAA1 has a redundant role to that of ATR/ ATRIP by binding to pRPA molecules. Prior studies have found that MVM infection inacti-vates the ATR pathway mediated by TOPBP1 [21]. In addition to preventing the activation of ATR signaling on host replication forks, we propose that depleting cellular stores of RPA mole-cules also depletes ETAA1 proteins from the host, thereby preventing ETAA1-mediated auto-regulatory ATR activation signals that protect the host genome. Since ETAA1 signaling functions in parallel to TOPBP1 to maintain genome stability, its absence renders the host cell sensitive to replication stress and formation of cellular DNA breaks. Future studies will eluci-date how ETAA1 and RPA molecules induce local signals on the MVM genome and how these interactions may modulate the virus life cycle.

Induction of host-cell replication stress by interfering with RPA function is a common pro-viral mechanism evolved by multiple viral families. The murine polyomavirus Large T Antigen associates with RPA using its origin-binding domain to sensitize host cells to DNA damage by chemicals and UV irradiation [40]. Similarly, infection with large DNA viruses such as Herpes Simplex Virus 1 (HSV1) induces activation of ATR and CHK1 that colocalize with ICP4 and ICP0 proteins in viral replication centers in the nucleus [41]. Consistent with these findings, the gamma-herpesvirus EBV recruits cellular DDR proteins in the homologous recombination repair pathway such as RPA, RAD51 and RAD52 to the newly synthesized EBV genome in the viral replication centers [42]. This recruitment of DDR proteins to herpesvirus genomes corre-lates with the induction of cellular DNA breaks in and around viral replication centers. The Vpr protein of HIV-1 and HIV-2 also induces replication fork stress on the host genome by inhibiting the cellular DNA repair pathways [43]. In a potentially similar manner, dependo-parvoviruses such as AAV, as well as the derivatives rAAV and UV-inactivated AAV, have been shown to induce replication stress in the host genome through the ATR/CHK1 pathway, possibly by mimicking stalled cellular replication forks [44]. In line with these observations, RPA has been shown to be associated with AAV DNA in nuclear foci and associated with AAV replication forks [45,46]. These findings invite the possibility that AAV and its modified

forms could induce global RPA exhaustion by depleting cellular RPA molecules, leading to replication stress in a similar manner to MVM. Therefore, RPA exhaustion seems to be an evolutionarily-conserved among viral pathogens to induce host replication stress by limiting ATR signaling as a means to induce host genome instability and facilitate virus replication.

Despite their inability to induce substantial levels of cellular DDR [22], ectopic expression of NS1 and NS2 generate replication stress on the host genome. This NS1-induced replication stress is dependent on its DNA binding function, as the K405S mutant of NS1 lacking its helicase and ATP-ase function does not induce replication stress [36,37]. ChIP-seq studies of ectopically expressed NS1 have previously found these molecules associated with cellular DDR sites and fragile genomic regions [8]. Therefore, we speculate that soluble NS1 proteins localize to pre-existing cellular DNA-break sites (many of which are fragile genomic sites) and amplify the genomic stress at these regions by perturbing the replisome at this milieu. However, the mechanism of how ectopic NS2 induces replication stress remains unknown. Since the requirement for NS2 in facilitating MVM replication is species specific, it is unlikely that NS2 plays a redundant role to that of NS1 in facilitating replication stress. MVM-NS2 interacts with CRM1 proteins to facilitate nuclear egress of the virus, likely mediated by interaction with the 14-3-3 subunits ε, β and/or ζ [9,47–49]. The 14-3-3 family of proteins regulate a wide variety of cellular processes, including apoptosis and cell cycle progression [50]. Out of these protein subtypes, the 14-3-3 variants γ, β, and ζ are known interactors of CHK1 [50]. Indeed, 14-3-3γ allows CHK1 to phosphorylate Cdc25A [51] and 14-3-3β/ζ binds CHK1 to retain CHK1 in the nucleus to promote cellular checkpoints [52]. We speculate that NS2-mediated sequestration of 14-3-3 γ, β, and ζ is a redundant pathway utilized by wild-type MVM during infection to inactivate CHK1 signaling in murine hosts. Future studies will elucidate the connection between these redundant cellular stress response pathways that are usurped by MVM and single-stranded DNA viruses for efficient pathogenesis. Taken together, our findings indicate that parvoviruses exploit the host cell's single-stranded DNA damage signaling machinery for their benefit to facilitate viral pathogenesis.

## Materials and methods

### Cell lines

Male murine A9 fibroblasts and female human U2OS osteosarcoma cells were maintained in Dulbecco's modified Eagle's medium (DMEM, high glucose; Gibco) supplemented with 5% Serum Plus (Sigma Aldrich) and 50 µg/ml gentamicin (Gibco). For synchronous infection experiments, cells were blocked in isoleucine deficient media as previously described [53]. Cells were cultured in 5% $CO_2$ at 37 degrees Celsius. Cell lines are routinely authenticated for mycoplasma contamination and background levels of DNA damage by γH2AX staining.

### Virus and viral infection

MVMp virus was produced in A9 and 324K cells as described previously [22]. MVMp infection was carried out at a Multiplicity of Infection (MOI) of 25 unless otherwise noted. MVM was irradiated in a UV-crosslinker (Stratagene) to generate UV-inactivated MVM virus as previously described [22].

### Plasmids, transfections, and inhibitors

RPA overexpression vector has been previously generated [38]. NS1, NS2 and NS1$^{K405S}$ overexpression vectors have been cloned into the pcDNA3.1 vector backbone and have been previously described [22]. All plasmids were transfected into A9 cells (0.5 micrograms) during

isoleucine deprivation at 36 hours post-synchronization using LipoD293 transfection reagent (SignaGen Laboratories). The MVMp4 oligo (Sequence: gataagcggttcagggagtttaaaccaaggcgc-gaaaaggaagtgggcgt; IDT) was transfected into A9 cells using the ThermoFisher RNAiMax transfection reagent according to manufacturer's protocol. CDC7 inhibitor PHA 767491 Hydrochloride was purchased from Millipore Sigma and used according to manufacturer's instructions. CDC7 inhibitor was used at final concentration of 5 µM and HU was used at a final concentration of 2 µM.

## DNA fiber assay

Murine A9 cells were synchronized in G0/G1 in Isoleucine deficient media, then released into complete media and transfected or infected according to experimental requirements [as described above and previously [53]]. Single molecule DNA Fiber Analysis experiments were performed according to established protocols [54]. At the end of the infection period, cells were pulsed in 20mM IdU for 20 minutes, immediately followed by pulsing with 50mM CldU for 20 minutes. Cells were then pelleted for 5 minutes at 5000xg and resuspended in 200 µL of complete media and stored on ice. 2 µL of cell solution was pipetted onto positively charged slides, then mixed with 6 µL of DNA Lysis Buffer and allowed to lyse for 5 minutes. Slides were then tilted so the fibers spread down the slide and allowed to air dry for 15 minutes. The DNA was then fixed to the slides with a 3:1 methanol:acetic acid solution. The DNA on the slides was denatured in 2.5 M HCl for 1 hour at room temperature before being blocked in 3% BSA in PBS for 30 minutes. After blocking, the cells were stained with Abcam rat anti-BrdU (1:1000) and BD Biosciences mouse anti-BrdU (1:500) at room temperature for 30 minutes, then washed with 0.1% Tween 20 in PBS 3 times and stained with anti-rat Alexa Fluor 488 and anti-mouse IgG1 Alexa Fluor 568 (1:1000) at room temperature for 30 minutes under covered conditions. Samples were washed with 0.1% Tween 20 in PBS 3 times and cover slips were affixed to slides using ProLong Gold Antifade Mountant (Thermo Scientific). Fibers were then imaged with a Leica Stellaris confocal microscope using a 63X oil immersion objective lens. Fiber lengths were measured using Digimizer software with the in-built "Path" function (Med-Calc Software Ltd). Categorization of the DNA fiber replication events were performed using the schematic illustrated in S3 Fig and as previously described [54].

## Antibodies

Antibodies used for DNA fiber analysis were: anti-BrdU (BD Biosciences, Clone B44, 347580), anti-BrdU (Abcam, ab6326), Alexa-Fluor 568 conjugated anti-mouse secondary (Thermo Scientific, A11004), Alexa-Fluor 488 conjugated anti-rat secondary (Thermo Scientific, A11006).

Antibodies used for western blot analysis were: Tubulin (Millipore Sigma, clone DM1A, 05–829), NS1 (2C9b monoclonal antibody), phospho-RPA32 (Cell Signaling, 83745), γH2AX (Abcam, ab11174), phospho-MCM2 (Thermo Fisher, PA5-106187), HRP-conjugated anti-mouse secondary (Cell Signaling, 7076S), HRP conjugated anti-rabbit secondary (Cell Signaling, 7074S).

Antibodies used for immunofluorescence analysis were: NS1 (2C9b monoclonal antibody), RPA32 (Cell Signaling, 2208S), phospho-RPA32 (Cell Signaling, 83745S), phospho-MCM2 (Thermo Fisher, PA5-106187), γH2AX (Abcam, ab11174), Alexa-Fluor 568 conjugated anti-mouse secondary (Thermo Scientific, A11004), Alexa Fluor 488 conjugated anti-rat secondary (Thermo Scientific, A11006), Alexa-Fluor 647 conjugated anti-rabbit secondary (Thermo Scientific, A21245).

Antibodies used for ChIP-qPCR analysis were: RPA (Cell Signaling, 2208S).

## Western blot analysis

Cell pellets were lysed on ice for 10 minutes in Radio-Immuno-Precipitation Assay (RIPA) buffer. The solid lysate was pelleted by centrifugation for 10 minutes at 5000xg at 4 degrees Celsius. Protein sample concentration was calculated using BCA assay (Bio-Rad) and equal amounts of lysate were loaded per well. The lysates were electrophoresed on relevant SDS-PAGE gels, transferred using a Biorad Trans-blot Turbo transfer system and monitored for the indicated levels of proteins using relevant primary antibodies, HRP-conjugated secondary antibodies and imaged on a LiCOR Odyssey imaging platform. The band intensities were quantified using the Integrated Density function on ImageJ.

## Chromatin Immunoprecipitation combined with Quantitative PCR (ChIP-qPCR)

MVMp-infected A9 cells were crosslinked in 1% Formaldehyde for 10 minutes at room temperature. The crosslinking reactions were quenched in 0.125 M glycine. Cells were lysed in ChIP lysis buffer (1% SDS, 10 mM EDTA, 50 mM Tris-HCl, pH 8, protease inhibitor) for 20 minutes on ice and the cell lysates were sonicated using a Diagenode Bioruptor Pico for 60 cycles (30 s on and 30 s off per cycle), before being incubated overnight at 4 degrees C with the antibodies bound to Protein A Dynabeads (Invitrogen). Samples were washed for 3 minutes each at 4 degrees C with low salt wash (0.01% SDS, 1% Triton X-100, 2 mM EDTA, 20 mM Tris-HCl pH8, 150 mM NaCl), high salt wash (0.01% SDS, 1% Triton X-100, 2 mM EDTA 20 mM Tris-HCl pH8, 500 mM NaCl), lithium chloride wash (0.25M LiCl, 1% NP40, 1% DOC, 1 mM EDTA, 10 mM Tris HCl pH8) and twice with TE buffer. DNA was eluted with SDS elution buffer (1% SDS, 0.1M sodium bicarbonate), crosslinks were reversed using 0.2M NaCl, Proteinase K (NEB) and incubated at 56 degrees C overnight. DNA was purified using PCR Purification Kit (Qiagen) and eluted in 100 µl of Buffer EB (Qiagen). ChIP DNA was quantified by qPCR analysis (Biorad) under the following conditions: 95˚C for 5 mins, 95˚C for 10 secs and 60˚C for 30 secs for 40 cycles. Interaction of RPA molecules with UV-MVM genome was assessed by qPCR assays using primers complementary to the MVMP4 promoter. The primer sequences in 5' to 3' direction are: Forward primer- TGATAAGCGGTTCAGGGAGT, Reverse primer- CCAGCCATGGTTAGTTGGTT.

## Immunofluorescence assays

Parasynchronized MVMp-infected A9 cells were grown on cover slips in round dishes. Cells were treated with CSK buffer for 3 minutes at room temperature, CSK buffer with Triton X-100 for 3minutes at room temperature and washed with PBS prior to fixation. Cells were crosslinked in 4% Paraformaldehyde in PBS (EMS; Gibco respectively) at room temperature for 10 minutes, washed in PBS and permeabilized in 1% Triton X-100 (in PBS) for 10 minutes. The samples were blocked in 5% BSA in PBS for 30 minutes, treated with the target antibodies (diluted in the blocking solution) for 30 minutes and then washed in PBS. Samples were treated with the target secondary antibody (diluted in blocking solution) for 30 minutes at room temperature. Samples were washed in PBS before being mounted on glass slides using DAPI Fluoromount (Southern Biotech). Samples were imaged on a Leica Stellaris5 Confocal Microscope using a 63X oil objective.

## Cell cycle analysis

Cell cycle analysis was performed by staining total DNA in the harvested cells with propidium iodide stain (Sigma Aldrich). Briefly, A9 cells were harvested, washed in 1 mL of PBS,

resuspended in 300 μl of PBS and fixed with 700 μl of chilled 100% Ethanol overnight at 4 degrees Celsius. Samples were resuspended in 300 μl of PBS, RNAse treated for 1 hour at 37 degrees Celsius and incubated with 5 μl of Propidium Iodide overnight. Cells were analyzed on a BD LSR Fortessa (University of Wisconsin-Flow Cytometry Laboratory) on the FL2 channel.

### Taqman qPCR analysis

Cells were harvested at the indicated timepoints, pelleted and resuspended in Cell Lysis Buffer (2% SDS, 0.15 M Sodium Chloride, 10 mM Tris pH 8, 1 mM EDTA). The lysed whole-cell extracts were treated with proteinase K (NEB) overnight at 37 degrees Celsius before genomic DNA was sheared with 25 G X 5/8 inch 1 mL needle syringe (BD Biosciences). MVM replication was measured using Taqman qPCR assays monitoring the formation of the plus strand using the following primers and probes: Fwd–agccgctgaacttggactaa, Rev–ctccttggtcaaggctgttc, Taqman probe–ccaaccatcccttaaaccct.

### Statistical analysis

Statistical analysis was performed using Graphpad Prism software. Comparison of the median values of IdU/CldU measurements from DFAs were performed by Mann Whitney Wilcoxon test as previously reported [55].

### Supporting information

**S1 Fig. Validation of cell synchronization of A9 mouse fibroblasts by Isoleucine deprivation.** A9 cells were plated in Isoleucine-deficient media for 36–42 hours (labelled as T0), before being released into complete DMEM media and infected (see Materials and Methods for details). Cells were harvested at 0 hours post release or 14 hours post release and processed for cell cycle analysis by propidium iodide staining. Cells enter S phase at 12 hours post-release, and are not impacted by the presence or absence of MVM (middle and right panels).
(TIFF)

**S2 Fig. Timecourse of MVM infection costained with NS1, RPA and γH2AX.** MVM infected A9 cells at 12 hpi (panel 1), 18 hpi (panel 2) and 24 hpi (panel 3) were processed for 4-color imaging. Nuclear borders were demarcated by white dashed lines and scale bars represent 10 microns.
(TIFF)

**S3 Fig. Schematic of categorization of MVM-induced replication forks.** Representative images of replication fibers in A9 cells that are (A) stalled, (B) new origin and (C) progressing, with their corresponding schematic for categorization shown on the right of the respective panel. Scale bars represent 4 micrometers.
(TIFF)

**S4 Fig. Validation of lack of NS2 produced by MVM$^{ΔNS2}$ virus.** (A) U2OS cells were infected with wild-type MVMp or MVM$^{ΔNS2}$ for 24 hours, and processed for NS1, NS2 and Tubulin levels by immunoblot. (B) Categorization of replication forks in U2OS cells infected with wild-type MVMp or MVM$^{ΔNS2}$ for 24 hours.
(TIFF)

**S5 Fig. Validation of UV-inactivation of MVM in A9 cells.** (A) Immunoblots showing inactivation of UV-MVM by monitoring NS1 levels during infection of A9 cells for 24 hours. Verification of lack of induction of cellular γH2AX upon infection of A9 cells with UV-MVM for 24 hpi. The respective band intensities are shown below the sample (B) Taqman qPCR analysis of MVMp with the probe directed against the plus strand of the MVM genome, showing

UV-MVM is made up of single stranded DNA at 18 hpi and therefore cannot be detected, ****
represents P $\leq$ 0.0001.
(TIFF)

**S6 Fig. Timecourse of MVM infection costained with NS1, RPA and phospho-RPA.** MVM
infected A9 cells at 12 hpi (panel 1), 18 hpi (panel 2) and 24 hpi (panel 3) were processed for
4-color imaging for the indicated proteins. Nuclear borders were demarcated by white dashed
lines and scale bars represent 10 microns.
(TIFF)

**S1 File. RPA Paper Information Revisions.** Supplemental.zip file of all DNA fiber assay mea-
surements.
(ZIP)

## Acknowledgments

The authors acknowledge expert technical guidance from Lauren Bunke (IMV), Ephraim Leigl
(IMV) and Marchel Hill (Palmenberg Lab; IMV); and members of the Majumder lab for criti-
cal reading of the manuscript. We acknowledge Dr. Alessandro Vindigni (Washington Uni-
versity in St. Louis), Dr. Annabelle Quinet (INSERM) and Dr. Kavi Mehta (Vanderbilt
University School of Medicine) for critical input on DNA Fiber Assays.

## Author Contributions

**Conceptualization:** MegAnn K. Haubold, Jessica N. Pita Aquino, Edward Pham, Kinjal
Majumder.

**Data curation:** MegAnn K. Haubold, Isabella K. Jones, Kinjal Majumder.

**Formal analysis:** MegAnn K. Haubold, Sarah R. Rubin, Isabella K. Jones, Clairine I. S. Larsen,
Edward Pham, Kinjal Majumder.

**Funding acquisition:** Kinjal Majumder.

**Investigation:** MegAnn K. Haubold, Jessica N. Pita Aquino, Sarah R. Rubin, Isabella K. Jones,
Clairine I. S. Larsen, Edward Pham, Kinjal Majumder.

**Methodology:** MegAnn K. Haubold, Clairine I. S. Larsen, Kinjal Majumder.

**Project administration:** Kinjal Majumder.

**Resources:** Kinjal Majumder.

**Supervision:** Kinjal Majumder.

**Validation:** Clairine I. S. Larsen.

**Visualization:** MegAnn K. Haubold, Jessica N. Pita Aquino, Sarah R. Rubin, Isabella K. Jones,
Kinjal Majumder.

**Writing – original draft:** MegAnn K. Haubold, Kinjal Majumder.

**Writing – review & editing:** MegAnn K. Haubold, Jessica N. Pita Aquino, Kinjal Majumder.

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
