## [Decision Letter · Decision Letter 0]

21 Mar 2023

Dear Dr. Majumder,

Thank you very much for submitting your manuscript "Genomes of the Autonomous Parvovirus Minute Virus of Mice Induce Replication Stress Through RPA Exhaustion" for consideration at PLOS Pathogens. As with all papers reviewed by the journal, your manuscript was reviewed by members of the editorial board and by several independent reviewers. In light of the reviews (below this email), we would like to invite the resubmission of a significantly-revised version that takes into account the reviewers' comments.

The editor strongly agrees with the requests of reviewers 1 and 3 to provide additional detail and necessary clarity on how the DNA fiber assays were done and analyzed. For example, to quantify stalled forks, did the authors quantify the number of red signals only, or did they use slower replication fork progression as a readout for stalled replication forks? If the latter is true the authors have to change the interpretation of their results. Similarly, are they considering only the green dots to define numbers of "new origins"? Without such and other detail, it cannot be evaluated whether the authors’ interpretations and conclusions are supported by their data. The other comments of these two reviewers will also need to be addressed. Some will require additional experimentation and the appropriate control experiments will need to be performed and included. This includes the request of reviewer 3 to perform experiments to investigate whether replicative MVM genomes also act as decoys for RPA. Finally, the first reviewer pointed out numerous examples of inconsistencies between text, figures, and figure legends. This lack of attention to detail in manuscript preparation is disappointing.

We cannot make any decision about publication until we have seen the revised manuscript and your response to the reviewers' comments. Your revised manuscript is also likely to be sent to reviewers for further evaluation.

Sincerely,

Karl Münger, Ph.D.

Academic Editor

PLOS Pathogens

Blossom Damania

Section Editor

PLOS Pathogens

Kasturi Haldar

Editor-in-Chief

PLOS Pathogens

orcid.org/0000-0001-5065-158X

Michael Malim

Editor-in-Chief

PLOS Pathogens

orcid.org/0000-0002-7699-2064

The editor strongly agrees with the requests of reviewers 1 and 3 to provide additional detail and necessary clarity on how the DNA fiber assays were done and analyzed. For example, to quantify stalled forks, did the authors quantify the number of red signals only, or did they use slower replication fork progression as a readout for stalled replication forks? If the latter is true the authors have to change the interpretation of their results. Similarly, are they considering only the green dots to define numbers of "new origins"? Without such and other detail, it cannot be evaluated whether the authors’ interpretations and conclusions are supported by their data. The other comments of these two reviewers will also need to be addressed. Some will require additional experimentation and the appropriate control experiments will need to be performed and included. This includes the request of reviewer 3 to perform experiments to investigate whether replicative MVM genomes also act as decoys for RPA. Finally, the first reviewer pointed out numerous examples of inconsistencies between text, figures, and figure legends. This lack of attention to detail in manuscript preparation is disappointing.

Reviewer's Responses to Questions

**Part I - Summary**

Reviewer #1: Employing single molecule DNA Fiber Analysis, Haubold et al. show that MVM infection induces the shortening of host DNA fibers as infection progresses and that the induction of replication stress is prior to the onset of MVM replication. Further, Haubold and colleagues demonstrate that the ectopic expression of the viral non-structural proteins, NS1 and NS2, can cause replication stress. However, as shortening of host DNA fibers was observed prior to expression of the non-structural proteins, the authors further determined that the mere presence of non-replicative MVM genomes (UV-MVM) is sufficient. Besides, the authors unveiled that MVM genomes might act as a sink for cellular RPA repositories, consequently causing replication stress and rendering the host genome even more vulnerable to additional DNA breaks. The work is both interesting and novel, but some concerns need to be addressed.

Reviewer #2: In this manuscript by Haubold et al., the authors investigate how an autonomous parvovirus, the minute virus of mice (MVM), induces host DNA damage. Majumder et al. have shown earlier that the emerging MVM replication centers are associated with sites of pre-existing cellular DNA damage. Here, they show that the shortening of the host´s replication forks and replication stress which enhances viral infection prelude viral replication. Specifically, the replication stress is induced by virus-mediated dysregulation of replication kinase CDC7 and the shortening of replication forks is guided by Viral NS proteins. They also suggest that association with the viral ss genome decreases the amount of nuclear replication protein A (RPA) and increases cellular DNA breaks.

The work presented in the manuscript is timely, important, and of high quality, thoughtfully addressing a variety of different hypotheses of how MVM infection leads to breaks in cellular DNA required for enhancement of the infection. I think this paper takes a rigorous, unbiased approach to the topic and will likely engage the journal's broad readership. I recommend publication after a few revisions.

Reviewer #3: In this manuscript, Haubold et al. investigated how MVM promotes replication stress caused by infection. Using DNA fibre assays, the authors show that viral infection with a replication potent or inactive form of the viral genome and over-expression of NS1 or NS2 lead to reduced replication fork progression. Replication stress induced by NBS1 is rescued when an ATPase mutant is used. Transfection of an expression plasmid for RPA32 in cells before infection rescue the replication stress induced by UV-inactivated MVM genomes. The authors also conclude that RPA accumulates on UV-inactivated MVM genomes by ChIP analyses performed with either an antibody against RPA or a control IgG. From these observations, the authors conclude that the MVM genome induced replication stress by acting as a decoy for the RPA molecule.

**Part II – Major Issues: Key Experiments Required for Acceptance**

Reviewer #1: 1. The authors performed single-molecule DNA Fiber Analysis (DFA) by IdU and CIdU pulsing in para-synchronized cells, showing a shortening of labelled DNA fibers upon MVM infection.

-the efficient synchronization should be shown

-Fig. 1B: HU was used as pos. ctrl. (HU-induced replication stress). However, no statistics is shown from the 2way ANOVA multiple comparison, even though the difference is significant, which is not that obvious by simply looking at the medians. The same holds true for Fig. 1C.

-The authors state in the figure 1 legend that statistical analysis for Fig. 1B and 1C was performed using Mann Whitney Wilcoxon test. However, statistical analysis in Fig. 1B was done by 2way ANOVA multiple comparison (see RPA Paper Information).

2. The authors categorized the DNA fiber types during the course of MVM infection, observing a large fraction of new origins at 24 hpi. As MVM infected cells at 24 hpi are supposed to be in G2 cell cycle phase where no new origins are expected to fire, the authors conclude that MVM modulates origin firing.

-how did the authors categorize the DNA fibers into progressing, stalled, and new origin?

-evidence should be provided that the MVM infected cells are indeed in G2 cell cycle phase at 24 hpi

3. The authors assessed the generalization of their observation by measuring the DNA fibers 24 hpi in non-synchronous U2OS cells infected with either wild-type MVM or MVM lacking NS2. NS2 is required for replication in murine but not in transformed human cells. They observed a DNA fiber shortening in either case, indicating replication stress induction at equivalent levels in U2OS cells.

-evidence for the lack of NS2 should be provided

-in the figure 1 legends, the authors write NS1-deficient mutant instead of NS2-deficient mutant (line 477)

-in the RPA Paper Information Fig. 1G is labeled as Fig 1I

-Axis label not consistent (B, C, G, H) and several data points are not within the range of the axis and are therefore not shown (compare to RPA Paper Information).

4. Treating MVM infected cells with an CDC7 inhibitor resulted in less DNA fiber shortening (significant for infected samples but not compared to mock) and in a decrease of new origins (Fig. 2B – 2D). Moreover, treating the MVM infected cells with the CDC7 inhibitor led to reduced levels of γH2AX (indicating less replication stress) and NS1 (less MVM replication; also seen in Fig. 2F; no mock-infection shown).

-efficient synchronization is not shown in the data

-Fig. 2B: not all data points are within the range of the axis and are therefore not shown (compare to RPA Paper Information Fig 2B.pzfx). 6 data points are outside the axis limit.

Fig. 2C: not all data points are within the range of the axis and are therefore not shown (compare to RPA Paper Information Fig 2C.pzfx). 8 data points are outside the axis limit.

-axis label not consistent (B, C).

5. Ectopically expressed MVM non-structural proteins NS1 and NS2

-were the cells synchronized for this experiment?

-is mock the pUC18 transfected sample?

-Fig. 3C: not all data points are within the range of the axis and are therefore not shown (compare to RPA Paper Information Fig 3C.pzfx). 1 data point is outside the axis limit.

-is there a synergistic effect of NS1 and NS2?

6. As replication stress was observable already early during infection (12 hpi) and prior to detectable NS1 protein expression, the authors concluded that NS1 alone might not be sufficient for MVM-mediated replication stress at early time points of infection and therefore assessed whether non-replicating MVM genomes lead to replication stress.

-Fig. 4C: not all data points are within the range of the axis and are therefore not shown (compare to RPA Paper Information Fig 4C.pzfx). 1 data point is outside the axis limit.

-Axis label not consistent (B, C)

-efficient UV-inactivation not confirmed (e.g., on protein level)

-UV-inactivated MVM is not supposed to induce γH2AX → based on ref.

-efficient synchronization is not shown

-the authors state that UV-MVM genome is largely made up of ssDNA. This should be shown at 12/18 hpi?

-is the result in Fig. 4E statistically significant?

-DFA analysis was performed at 18 hpi. In contrast, ChIP-qPCR was performed at 24 hpi and IF analysis at 16 hpi. Why did the authors use different times of infection?

-line 526-528: At 24 hours post-transfection,….. → There is no transfection in this experiment.

7. To assess whether RPA depletion results in MVM-induced replication stress, the authors overexpress pRPA, thereby showing that this can rescue the DNA fiber shortening upon UV-MVM infection.

-Fig. 5B: not all data points are within the range of the axis and are therefore not shown (compare to RPA Paper Information Fig 5B.pzfx). 16 data points are outside the axis limit.

-is Mock ctrl. transfected?

-is transfection itself the reason for the fiber shortening in the pRPA transfection? Missing comparison to ctrl. transfection.

-Fig. 5C: not all data points are within the range of the axis and are therefore not shown (compare to RPA Paper Information Fig 5C.pzfx). 4 data points are outside the axis limit.

-axis label not consistent (B, C)

Reviewer #2: Fig. 4F. The data would be more convincing after quantitative colocalization or closeness

Analyses. At least instead of the term colocalization use the phrase localized within (line:227.)

Figures with western blots. Please add quantitative analyses of WBs.

Reviewer #3: The current version of the manuscript lacks important details about how the DNA fibre analyses were done, thereby preventing the interpretation of the data and the conclusions made by the authors. The authors present data potentially supporting a role of the viral genome, the viral protein NS1 and NS2, in promoting replication stress. Without proper control experiments and further characterization of these phenotypes, the current data do not support the conclusions made by the authors.

Major points to be revised:

- Provide a detailed protocol on how the DNA fibre assays were quantified and images for the phenotype considered stalled replication forks vs new origin firing.

- In figure 1E, the analysis of the phosphorylation of H2AX by western blot is not robust (as shown by the major difference between �-H2Ax levels obtained in Fig.1E vs Fig. 2E) �-H2Ax levels should be evaluated by immunofluorescence, which is more sensitive and more robust. Co-investigation of �-H2Ax levels and pRPA would also be essential to conclude that DNA damage signalling is induced after replication stress in MVM-infected cells.

- It is complicated to investigate if the replicative MVM genome also acts as a decoy for RPA. However, validating the hypothesis that ssDNA is leading to this phenotype can be done by investigating if the transfection of scrambled ssDNA also led to the same phenotype or if an MVM genome containing a mutation in the NS1 gene (should also inactivate replication) led to a similar phenotype.

Proper controls to be added:

- Cytometry analysis of the cell cycle should be provided to show that isoleucine depletion lead to synchronization of cells in G0/G1 and cell cycle progression in S and G2 phases after the addition of complete media.

- In Fig 2, the authors should show that CDC7 is inhibited in iCDC7-treated cells. This could be done by doing a western blot against CDC7 or phosphorylated MCM2 as described previously (Montagnoli et al. (2008) Nature Chemical biology).

- Validation of MVM genome inactivation should be provided.

**Part III – Minor Issues: Editorial and Data Presentation Modifications**

Reviewer #1: (No Response)

Reviewer #2: General remarks:

Was there any specific reason why the IdU was not introduced prior to infection?

The colors used in the figures might not be visible for people with color vision deficiencies, I would suggest changing the color schemes in the figures.

1) The last sentence in the abstract should be rewritten and clarified.

2) Line 74: nuclear export

3) Line 86: jumpstart ;)

4) Figure 1A/B: is there a statistically significant change between mock and 12 and 18 hpi and

HU. Please show the analyses in the Figure. Fig. 1E Tublin = tubulin.

5) Fig 1G and 1H: Is the effect of NS2-expression mutant infection on replication fiber progress

(stalling and emergence of new origins) similar to that of wt MVM infection?

6) Add spaces between the result chapters

7) Fig 3 B/C. It would have been very interesting to see if there is a statistically significant

difference between wt MVM and ectopic expression of NS1 and NS2 (if that is possible to

include in the figure).

8) Line 208. Could you please explain more carefully what is the time schedule of the MVM

NS1 expression.

9) Lines: 182-184 and 240-241 Can this be said merely based on the increase of NS1?

Reviewer #3: Minor comments

- The concept of DDR vs DNA damage signalling and repair should be revisited in the whole manuscript. DNA damage response (DDR) refers to the signalling induced to stop the progression of the cell cycle upon detection of DNA damage.

- Detailed drug treatments (HU, iCDC7) should be added to the manuscript.

PLOS authors have the option to publish the peer review history of their article (what does this mean?). If published, this will include your full peer review and any attached files.

Reviewer #1: No

Reviewer #2: No

Reviewer #3: No
---

## [Editor Report · Decision Letter 1]

18 May 2023

Dear Dr. Majumder,

We are pleased to inform you that your manuscript 'Genomes of the Autonomous Parvovirus Minute Virus of Mice Induce Replication Stress Through RPA Exhaustion' has been provisionally accepted for publication in PLOS Pathogens.

Best regards,

Karl Münger, Ph.D.

Academic Editor

PLOS Pathogens

Blossom Damania

Section Editor

PLOS Pathogens

Kasturi Haldar

Editor-in-Chief

PLOS Pathogens

orcid.org/0000-0001-5065-158X

Michael Malim

Editor-in-Chief

PLOS Pathogens

orcid.org/0000-0002-7699-2064
---

## [Editor Report · Acceptance letter]

25 May 2023

Dear Dr. Majumder,

We are delighted to inform you that your manuscript, "Genomes of the Autonomous Parvovirus Minute Virus of Mice Induce Replication Stress Through RPA Exhaustion," has been formally accepted for publication in PLOS Pathogens.

Best regards,

Kasturi Haldar

Editor-in-Chief

PLOS Pathogens

orcid.org/0000-0001-5065-158X

Michael Malim

Editor-in-Chief

PLOS Pathogens

orcid.org/0000-0002-7699-2064